

# Artificial light source selection in seaweed production: growth of seaweed and biosynthesis of photosynthetic pigments and soluble protein

Shitao Huang[1,*], Ke Li[1,*], Yaoru Pan[1], Yan Yu[1], Thomas Wernberg[2], Thibaut de Bettignies[2,3], Jiaping Wu[1], Chaosheng Zhou[4], Zhixing Huang[4] and Xi Xiao[1]

[1] Ocean College, Zhejiang University, Zhoushan, Zhejiang, China

[2] UWA Oceans Institute and School of Plant Biology, University of Western Australia, Crawley, Western Australia, Australia

[3] UMS 2006 Patrimoine Naturel (PatriNat), OFB-CNRS-MNHN, Muséum national d'Histoire naturelle, Paris, France

[4] Zhejiang Mariculture Research Institute (Zhejiang Key Lab of Exploitation and Preservation of Coastal Bio-resource), Wenzhou, Zhejiang, China

[*] These authors contributed equally to this work.

## ABSTRACT

Seaweed growth is often limited by light. Artificial light supply has been well studied in terrestrial agriculture, however, much less is known about its effect in seaweed aquaculture. In this study, the effects of four artificial light sources (white, red, green, and blue LEDs light) on a brown alga *Sargassum fusiforme* and a green alga *Ulva pertusa* were investigated. Seaweed growth, accumulation of photosynthetic pigments (chlorophyll *a* and carotenoid), and soluble protein were evaluated. White LED light was the optimal supplementary light when cultivating *Ulva pertusa* and *Sargassum fusiforme*, because it promoted seaweed growth while maintaining protein production. Meanwhile, red LED was unfavored in the cultivation of *S. fusiforme*, as it affected the seaweed growth and has a lower residual energy ratio underneath the water. LEDs would be a promising supplementary light source for seaweed cultivation.

## INTRODUCTION

The coastal ecosystem provides a variety of ecosystem goods and services that support the sustainable development for human beings (*Bennett et al., 2016*; *Mehvar et al., 2018*; *Wu et al., 2020*). Among them, seaweeds cover a large area of the coastal zone, providing high-value ecosystem services (i.e., globally significant carbon fixation, absorb contaminants, etc.) and raw material for food, fertilizer, and pharmaceutical industries (*Capuzzo et al., 2015*; *Duarte et al., 2017*; *Xiao et al., 2017*; *Xiao et al., 2019a*; *Xiao et al., 2021*; *Pan et al., 2018*). However, human activities and global climate change are currently posing a high pressure on the coastal ecosystems (*Xiao et al., 2015*; *Zhao et al., 2017*; *Smale et al., 2019*; *Huang et al., 2020*; *Huang et al., 2021*; *Tang et al., 2021*). Natural seaweeds are facing the

Corresponding author
Xi Xiao, xi@zju.edu.cn

threat of ecological degradation (*Xiao et al., 2019b*; *Xiao, Huang & Holmer, 2020*), and there is an increasing demand for large-scale seaweed aquaculture (*Xiao et al., 2017*).

Nevertheless, problems such as warming, high sediment load, epiphyte cover, disease, and fish grazing seriously affect the development of large-scale seaweed farming (*Ateweberhan, Rougier & Rakotomahazo, 2015*). A fundamental factor affecting seaweed growth is light limitation (*Xiao et al., 2019a*). Modern mariculture with excessive fertilizer application retains a large number of nutrients and contaminants in seaweed cultivation area, increasing the turbidity of seawater (*Lu, Wang & Feng, 2017*). Light availability, which is limited by water transparency, directly determines the photosynthesis activity of seaweeds and their biosynthesis ability, causing ecological and economic loss to seaweed farms (*Orfanidis, 1992*). For instance, Zhoushan Island in the East China Sea, situated at the mouth of the Yangzi River, is encountering turbid water, and seaweed cannot grow well in such coastal waters (*Tseng, 1987*). Light limitation may therefore also restrict the important ecological functions of seaweed farming, such as nitrogen and phosphorus removal (*Matos et al., 2006*; *Abreu et al., 2011*; *Xiao et al., 2017*; *Xiao et al., 2019a*). Hence, artificial lighting may be a solution to encourage seaweed growth under a light-limitation situation (*Xiao et al., 2017*; *Xiao et al., 2019a*).

Light-emitting diodes (LEDs) produce monochromatic light in an energy-efficient way, suggesting their potential to provide supplementary light for seaweed growth (*Bourget, 2008*; *Kim et al., 2015*). By filtering fluorescent light with band-pass filters, monochromatic lights have been produced to promote seaweed growth, and their influence has been tested on several seaweed species (*Figueroa & Niell, 1990*; *Figueroa et al., 1995*; *Korbee, Figueroa & Aguilera, 2005*; *Kim et al., 2015*; *Bonomi Barufi, Figueroa & Plastino, 2015*). In general, compared to fluorescent light culturing, the seaweed growth rate could be increased by 10–60% under suitable LED light conditions (*Schulze et al., 2016*; *Kim, Choi & Lee, 2019*; *Gong, Liu & Zou, 2020*; *Öztaşkent & Ak, 2020*). However, although LED light has been proposed as a light source for *Gracilaria* cultivation (*Kim et al., 2015*; *Bonomi Barufi, Figueroa & Plastino, 2015*), its influence on a broader variety of seaweed species and their biosynthesis remains poorly understood. Besides, different wavelengths of lights vary in their ability to penetrate water. For instance, in general, red light is most likely to be absorbed by water, and thus blue and green light can go deeper than red light (*Chiang, Chen & Chen, 2011*). Therefore, there is an urgent need to further test the influence of LED light on the growth and biosynthesis of seaweed.

In this experiment, white, blue, green and red LED light were tested as artificial light sources to support the cultivation of two common and economically important seaweed - *Ulva pertusa* and *Sargassum fusiforme*. We assessed the seaweed growth, photosynthetic pigments, and soluble protein accumulation. Biomass accumulation and lighting-harvesting efficiency are important factors in seaweed cultivation. Meanwhile, seaweed proteins are essential for their biological processes and may become important food sources (*Cai et al., 2005*; *Postma et al., 2018*). Our results will facilitate the selection of supplementary artificial light sources for seaweed cultivation.

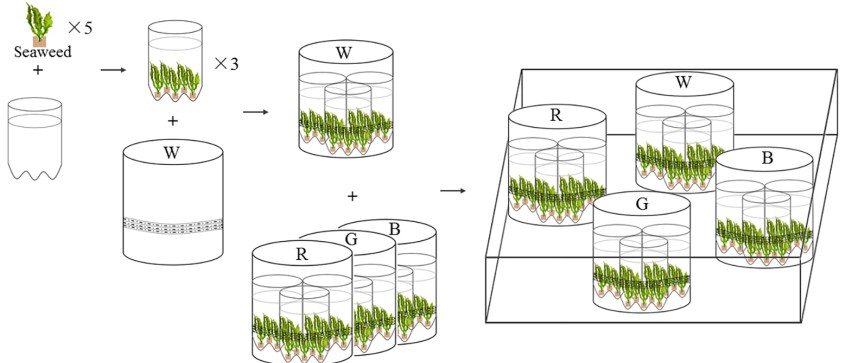

**Figure 1** **Experimental settings of LED culture system for _U. pertusa_ and _S. fusiforme._** W, R, G, B represent white, red, green, blue LED light treatments. The LED lighting system was assembled in white PVC tubes (height: 25 cm, diameter: 20 cm), with flexible rope LED lights (rope length is 3 m) affixed to the inner walls of each PVC tubes.

## MATERIALS & METHODS

### Seaweed species and cultivation

The juveniles of two seaweed species, _Sargassum fusiforme,_ and _Ulva pertusa,_ were collected from Dongtou Island, Wenzhou City, Zhejiang Province, China (27°41′42″ N, 1211°1′06″E). For acclimation, all the collected seaweeds were maintained in glass jars containing filtered, sterile natural seawater (33‰), and the temperature of seawater was controlled at 16 °C by a chiller (LS16-600, JLLN, Shenzhen, China). Illumination was provided by fluorescent lamps (120 µmol photons $m^{-2}s^{-1}$, 12 h of light followed by 12 h of darkness). After 3 days of acclimation, healthy individual seaweeds were selected and used in the following experiments.

### Light sources

Four LEDs emitting white, red, green, and blue light were used as artificial light sources for seaweed cultivation, providing 120 µmol photons $m^{-2}s^{-1}$ on the surface of seaweed thalli. The LED lighting system was assembled in PVC tubes (height: 25 cm, diameter: 20 cm), with flexible LED light belts (length: 3 m) affixed to the inner walls of each PVC tubes (Fig. 1). All the LED lighting diodes (Opple Co. Ltd., Shanghai, China) were driven by a 220V power supply. The light was supplied for 12 h every day from 6:00 a.m to 6:00 p.m. Light spectra were measured with an optical spectrum analyzer (CMS-2S, Inventfine Co. Ltd., Hangzhou, China). Light intensity was measured using an MQ-200 Quantum Separate Sensor (Apogee Instruments, USA). Residual energy ratio (_Rer_) underwater was measured following Chiang's method (Table S2) (_Chiang, Chen & Chen, 2011_). Among the four LED lights, the red LED light has the lowest _Rer_ value, followed by the white, green, and blue light, respectively (Table S2).

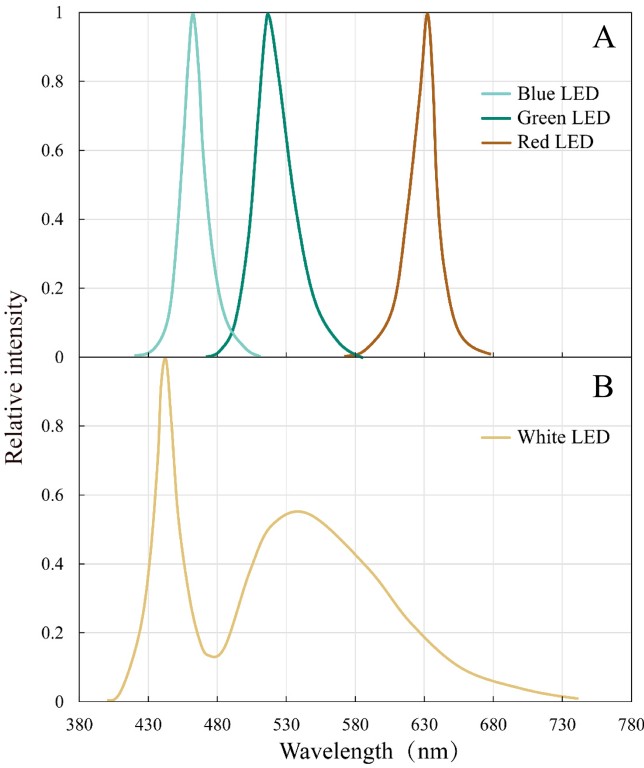

**Figure 2** Emission spectral distribution of the the white, red, green and blue LEDs light sources. (A) Red, green, blue LEDs (B) White LEDs.

## Emission spectral distribution of light sources

The peak wavelengths of the red, green, and blue LEDs were 632 nm, 517 nm, and 462 nm, respectively, and all the peaks are narrow (80–100 nm) (Fig. 2). White LED had a continuous spectrum with two peaks in the blue and green light region.

## Light incubation experiments

The cultivation lasted for 18 days. Five juvenile individual seaweeds (approx. 5 g fresh weight) were placed into one cylindrical plastic bottle (1000 ml, diameter: 100 mm). Three replicate bottles were placed inside the PVC tubes for light treatments (Fig. 1) (*Kim et al., 2015*). In total, 15 juveniles of *U. pertus* and 12 juveniles of *S. fusiforme* were set in a PVC tube. The LED lights were controlled independently. During the experimental period, both *S. fusiforme* and *U. pertusa* were cultivated in filtered and sterile natural seawater. The nutrients, phosphate ($PO_4^{3-}$) and nitrate ($NO_3^-$) were renewed every second day. The $PO_4^{3-}$ and $NO_3^-$ concentrations of seawater were 15 $\mu$mol $L^{-1}$ and 150 $\mu$mol $L^{-1}$, respectively. The seawater was sufficiently aerated by an air pump. The temperature in the aquarium was kept at 16 °C.

## Growth

Specific growth rate (SGR) was calculated following the method used in our previous study (*Xiao et al., 2015*):

$$SGR = \ln\left(\frac{W_t}{W_0}\right) \times t^{-1} \times 100 \qquad (1)$$

where $W_0$ is the initial algal biomass, and $W_t$ is the algal biomass after $t$ days of cultivation. Fresh weights (FW) of *S. fusiforme* and *U. pertusa* were measured every second day.

## Photosynthetic pigment and soluble protein

The photosynthetic pigment Chlorophyll *a* and carotenoid content were measured. Chlorophyll *a* was extracted using acetone (90%) neutralized with sodium carbonate, as described in *Jeffrey & Humphrey (1975)*. Carotenoid concentrations were detected following Seely, Duncan, and Vidaver (*Seely, Duncan & Vidaver, 1972*). Chlorophyll *a* and carotenoid were measured every second day by spectrophotometer (Inesa 722S, Shanghai, China). The soluble protein concentrations were also determined spectrophotometrically at 595 nm by Coomassie brilliant blue method every second day (*Bradford, 1976*).

## Data analysis

Differences between light treatments were tested for each species separately using one-way ANOVA with a significant level of $p < 0.05$. Tukey's test was used for the post hoc test. Data normality was checked by the shapiro-wilk test and variances homogeneity were checked by Levene's test. Differences between data that did not pass the normality test were analyzed by Kruskal–Wallis $H$ test (non-parametric). Data with differences that did not pass Levene's test were analyzed by Welch's ANOVA, where the Games-Howell test was used for post hoc test. Statistical tests were performed with SPSS (version 19.0).

# RESULTS

## Specific growth rate

For both seaweed species *(Ulva pertusa* and *Sargassum fusiforme)*, white-LED light stimulated seaweed growth and the light colors of LEDs differed in their influences on seaweed growth. The white light stimulated the growth of *U. pertusa* (average SGR = $5.16 \pm 0.88\%\ d^{-1}$) significantly, compared to the green light ($4.07 \pm 0.64\%\ d^{-1}$) (Fig. 3, $p < 0.05$). As for *S. fusiforme*, SGR decreased following the sequence of white LED light ($3.21 \pm 1.10\%\ d^{-1}$) >green and blue LED light ($2.44 \pm 1.13$ and $2.35 \pm 0.66\%\ d^{-1}$, $p > 0.05$) >red LED light ($1.34 \pm 0.39\%\ d^{-1}$, $p < 0.05$).

## Photosynthetic pigments and soluble protein

The LED lights induced changes in pigments and soluble protein synthesis in the two seaweed species. For instance, the Chl *a* concentration of *U. pertusa* treated with red LED light ($1.21 \pm 0.15\ mg\ g^{-1}$) was higher than those treated with white ($0.92 \pm 0.19\ mg\ g^{-1}$, adjusted $p = 0.004$, $p = 0.001$), green ($1.00 \pm 0.20\ mg\ g^{-1}$, adjusted $p = 0.040$, $p = 0.007$) and blue LED lights ($1.06 \pm 0.13\ mg\ g^{-1}$, adjusted $p = 0.491$, $p = 0.082$) (Fig. 4). For *S. fusiforme*, Chl *a* concentration of the seaweed were the same in all the light treatments.
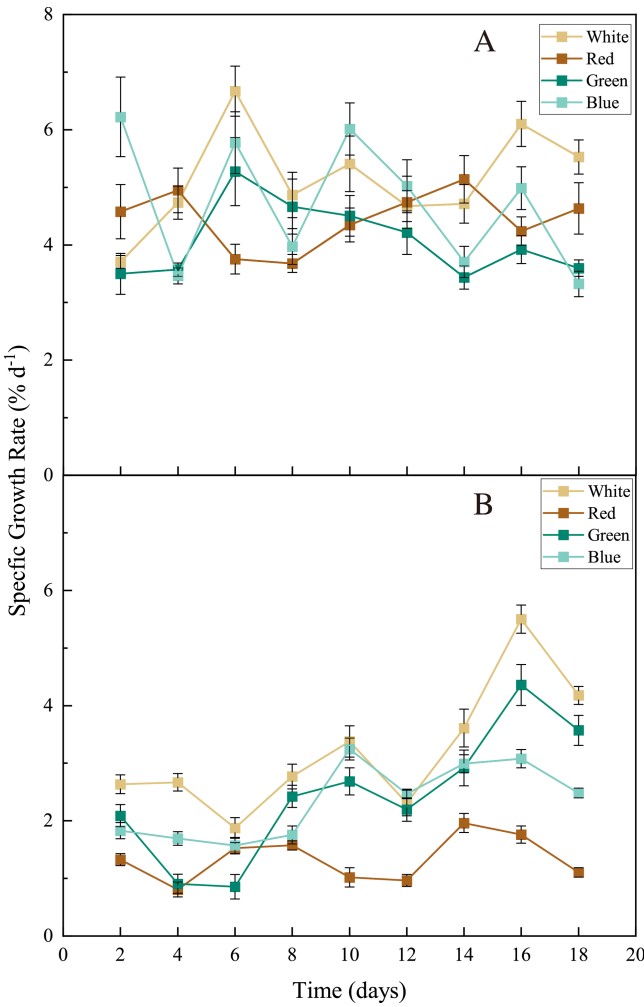

**Figure 3** Specific growth rate (SGR) of *U. pertusa.* and *S. fusiforme* after 18 days cultivation under various LEDs. (A) SGR of *U. pertusa* (B) SGR of *S. fusiforme.*

Carotenoid concentrations shared similar levels in different LED groups for both *U. pertusa* and *S. fusiforme* ($p > 0.05$). The concentrations of soluble protein showed no significant difference among four LED groups in *U. pertusa* and *S. fusiforme* (Fig. 5).

## DISCUSSION

### Light driven shifts in seaweed growth

For both seaweed species *U. pertusa* (green algae) and *S. fusiforme* (brown algae), the experimental seedlings achieved the highest growth rate under white LED lighting, which is consistent with previous studies (*Tovar et al., 2000*; *Kim et al., 2015*). This may be partially explained by the broad light wavelengths of white light (from 430 nm to 630 nm) (Fig. 2). White LED light, with the ability to provide spectrum comparable of sunlight (*Glemser et al., 2016*), is capable of supporting C and N metabolism in seaweeds (*Tsekos et al., 2002*; *Korbee, Figueroa & Aguilera, 2005*). White LED light with a broad continuous emission

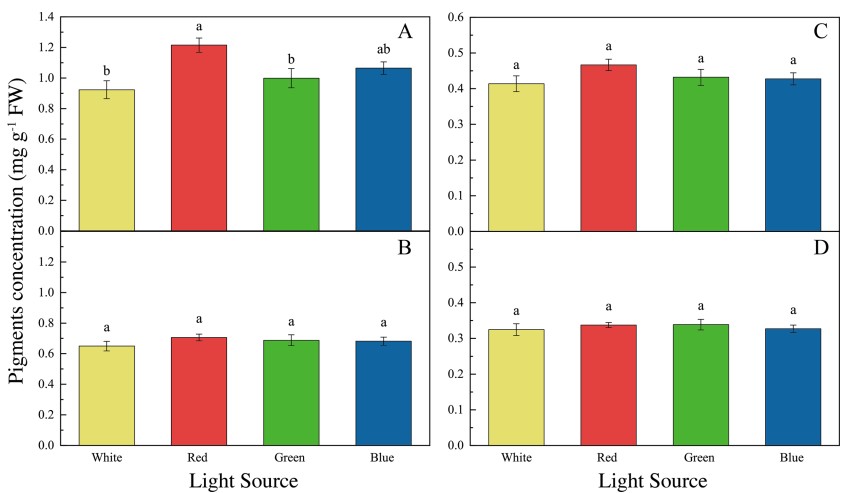

**Figure 4** **Pigments concentrations of *U. pertusa* and *S. fusiforme* after 18 days cultivation under various LEDs.** (A) Chlorophyll *a* concentration of *U. pertusa* (B) Carotenoid concentration of *U. pertusa* (C) Chlorophyll *a* concentration of *S. fusiforme* (D) Carotenoid concentration of *S. fusiforme*.

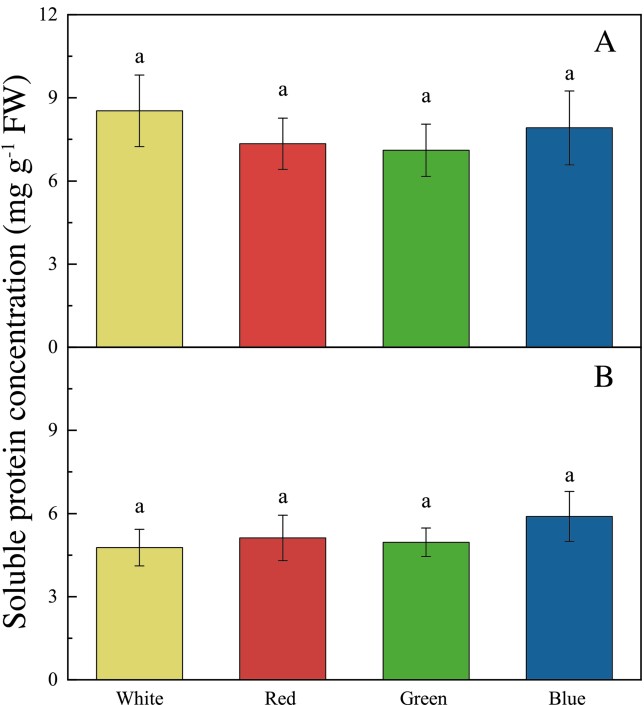

**Figure 5** **Soluble protein concentrations of *U. pertusa* and *S. fusiforme* after 18 days cultivation under various LEDs.** (A) Soluble protein concentration of *U. pertusa* (B) Soluble protein concentration of *S. fusiform*.

spectrum, is also providing a higher luminous efficiency compared to a fluorescent white-light source (*Pimputkar et al., 2009*). *S. fusiforme* cultivated under red LED lights had a significantly lower growth rate compared to those treated with white LED lights. *S. fusiforme* contains fucoxanthin, one of the brown algae carotenoids that plays an important role in photosynthesis (*Terasaki et al., 2005*; *Xiao et al., 2012*). The main absorption peaks of fucoxanthin are in the blue region (about 455 nm) (*Wang et al., 2005*). Thus, red light is not likely to be utilized at high efficiency by *S. fusiforme*. Another species of brown algae *Sargassum horneri* also showed slower growth than the individuals cultivated in white or blue LED lights (*Miki et al., 2017*). Interestingly, red LED light has been widely applied in the cultivation of microalgae and terrestrial plants (*Goins et al., 1997*; *Poudel, Kataoka & Mochioka, 2008*) (see also Table S1). However, a negative influence of red LED light on *S. fusiforme* growth was found in our experiment. This hinted again, the importance to further investigate more seaweed species since the influence of light quality appeared to be highly species-dependent.

Through the 18 days of cultivation, *U. pertusa* seemed to maintain a relatively fast and stable growth under all the LED lights with constant growth rates, while *S. fusiforme* held a higher growth rate in the last days than the earlier days (Fig. 3). *Ulva* species (i.e., *U. lactuca* and *U. prolifera*) grow faster than many other macroalgal species (*Pedersen, Borum & Fotel, 2010*; *Tang et al., 2021*). Also, the simple morphology of *U. pertusa* leads to an easier adaptation to the environment (including changes in the light condition). On the contrary, *S. fusiforme* may take a longer time for adaptation when cultured in changed light conditions (Fig. 3). Another brown alga *Sargassum horneri*, which was morphologically similar to *S. fusiforme*, showed a similar growth pattern when cultured in LED lights (*Miki et al., 2017*). The thalli from *S. horneri* were found to absorb red light in low efficiency, we suspect that the utilization rate of red light may also be lower for *S. fusiforme* thalli, leading to a lower growth rate under red LED lights (*Matsui, Ohgai & Murase, 1994*; *Miki et al., 2017*).

## Light-driven shifts in seaweed biosynthesis

The accumulation of photosynthetic pigments and soluble protein in *U. pertusa* and *S. fusiforme* were also influenced by light sources. Chl *a* concentration was significantly higher in *U. pertusa* under red LED light as compared to white and green LED lights. Similar to our findings, several other green algae *Ulva prolifera* and *Ulva lactuca* held higher Chl *a* content in red LED cultivation compared to white and blue LED lights (*Takada et al., 2011*; *Gong, Liu & Zou, 2020*). However, *U. pertusa* was found to synthesize less Chl *a* and form smaller chloroplast under red light, as compared to blue and white lights (*Muthuvelan, Noro & Nakamura, 2002*; *Le et al., 2018*). The higher Chl *a* concentration per fresh weight of *U. pertusa* may be derived from the restrained biomass accumulation under red light.

As for carotenoid and soluble protein content, no significant difference was found among the four light treatments for both *S. fusiforme* and *U. pertusa* in our study. Previously, the red light was found to promote carotenoid synthesis in a green algae *Dunaliella salina* to reduce reactive oxygen species formation and increase anti-oxidant level (*Xu & Harvey, 2019a*; *Xu & Harvey, 2019b*).

### Seaweed cultivation in fields using supplementary LED lights

Seaweeds play an important role in food and feed supply (*Makkar et al., 2016*). LED lights could stimulate the growth and increase the yield of specific seaweeds as compared to traditional fluorescent light (*Kim, Choi & Lee, 2019*; *Gong, Liu & Zou, 2020*). The growth and biochemical composition of seaweed were affected by the light quality, indicating the potential for using artificial light to increase the yield and proportion of high-value biomolecules in seaweed aquaculture (Table S1). There are plenty of commercially available LED devices, and underwater LED lighting has been developed for many years, which makes LED cultivation systems easy to be established for both land and offshore seaweed cultivation (*Hardy et al., 2008*; *Shen et al., 2013*). Our indoor experiment showed that white LEDs were favored in the cultivation of *U. pertusa* and *S. fusiforme* because white LED promoted seaweed growth while protein production was maintained at a constant level. White LED has also a relatively good ability to penetrate underwater (Table S2). Nevertheless, the turbidity of water is still an important factor to be considered in fields. For seaweed cultivation on lands, water renewal or flow water are usually applied, so it is relatively easy to keep the water clean. But for seaweed growing in a natural water body, such as for seaweed-based ecosystem restoration, turbidity and other environmental factors need to be considered in the future study.

## CONCLUSIONS

To summarize, this investigation highlighted the potential of the supplementary LED light source in seaweed cultivation. The results indicated that the effects of artificial light on seaweed, including the growth rate, photosynthetic pigments, and soluble protein are highly species-dependent. Therefore, we propose that manipulating the artificial light source for seaweed research and seedling industries is a promising venture.

## ACKNOWLEDGEMENTS

We thank Peng Zhang and Yining Zhang at Marine Aquaculture Research Institute of Zhejiang Province for guidance in seaweed cultivation.

### Funding

This study was supported by the Major Science and Technology Program for Water Pollution Control and Treatment (2018ZX07208-009), the National Natural Science Foundation of China (grant no. 21876148 & 21677122), and the International Science & Technology Cooperation Program of China (grant no. 2015DFS01410). Thomas Wernberg was supported by the Australian Research Council (DP160100114). The funders had no role in study design, data collection and analysis, decision to publish, or preparation of the manuscript.

## Grant Disclosures

The following grant information was disclosed by the authors:

Major Science and Technology Program for Water Pollution Control and Treatment: 2018ZX07208-009.

National Natural Science Foundation of China: 21876148, 21677122.

International Science & Technology Cooperation Program of China: 2015DFS01410.

Australian Research Council: DP160100114.

## Competing Interests

The authors declare there are no competing interests.

## Author Contributions

- Shitao Huang analyzed the data, prepared figures and/or tables, and approved the final draft.
- Ke Li conceived and designed the experiments, performed the experiments, analyzed the data, prepared figures and/or tables, and approved the final draft.
- Yaoru Pan, Jiaping Wu, Chaosheng Zhou and Zhi Xing Huang analyzed the data, authored or reviewed drafts of the paper, and approved the final draft.
- Yan Yu performed the experiments, analyzed the data, authored or reviewed drafts of the paper, and approved the final draft.
- Thomas Wernberg and Thibaut de Bettignies conceived and designed the experiments, analyzed the data, authored or reviewed drafts of the paper, and approved the final draft.
- Xi Xiao conceived and designed the experiments, performed the experiments, analyzed the data, authored or reviewed drafts of the paper, and approved the final draft.

## Data Availability

The raw measurements are provided in the Supplementary File.

## Supplemental Information

Supplemental information for this article can be found online at http://dx.doi.org/10.7717/peerj.11351#supplemental-information.

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
