# Peer review of "Artificial light source selection in seaweed production: growth of seaweed and biosynthesis of photosynthetic pigments and soluble protein"

_PeerJ, doi:10.7717/peerj.11351_

## Round 0.1 · original submission · Major Revisions

Having looked at the reviewers comments and read your paper I think it needs a bit of work before it is publishable. I may be missing something but don't see how the model adds to the fundamental and interesting result you have that LEDs can stimulate algal growth and that different coloured LEDs may impact growth rates. I agree with reviewer one in that it would be interesting to see something on the relative energy costs per g algae cultivated using LEDs versus fluorescent tubes.

You should present your data analysis more fully with test degrees of freedom, statistics and p values. The simplest approach might b to take use measurements from a few points during the experiment and use ANOVA with posthoc tests. Or you could of course use ANCOVA?

I have made some suggestions for grammatical corrections on the attached pdf.

·

Basic reporting

- Overall a worthwhile study investigating early stage, controllable and optimised growth of a sustainable aquaculture taxon which has strong cultural relevance to S.E. Asia.

- English is generally good but some minor care with plurals/use (or no use) of "the" - taking the first Introduction paragraph as an example. Occasionally, text slips from past into present tense, and some attention required on reference style (e.g. lines 124-125).

- Text would benefit from a little more explanation in the Introductory background, for instance line 46, what nutrients and why do they not support wild seaweed growth? Is this a case of contamination by heavy metals, for example? It would also be worth briefly explaining why you have chosen the particular biochemical fractions to sample, what do they do and how might they relate to each other (might sit best in lines 72-78)?

- Literature references appear fairly sound and it is unsurprising that much will be taken from microalgae. However a cursory glance suggested at least one recent reference below, which seems very relevant to this study:

Enhanced growth rate and ulvan yield of Ulva pertusa using light-emitting diodes (LEDs)
By: Le, Bao; Shin, Jong-Am; Kang, Man-Gu; et al.
AQUACULTURE INTERNATIONAL Volume: 26 Issue: 4 Pages: 937-946 Published: AUG 2018

Figure 1 (part) showing the lights is probably unnecessary, and a diagram rather than a photograph of the culture materials would be easier to follow. The latter is required as the written explanation is a little unclear (for instance, rope length does not equal the tube length, (line 91-92).

Figures 3-5 may be better viewed as colour relevant to treatment, but at least "open" rather than a block of black colour. It would also be an improvement to display as a series of simple tables.

Analysis - needs to be clearer as I would have thought that the results would all be ANOVA, so state why and when t test was used. Did data pass normality and variance tests so they were OK to enter a parametric test. There should also be details of the post hoc tests used. More P values should enter the results text, for example in brackets along with the general data values that have been picked out.

Discussion - OK overviews or results with reference to other papers - however is it possible to discuss more about potential reasons for your findings? Perhaps a bit basic, but with one brown alga and one green alga used, is it particularly surprising that the wavelengths at the same parts of the colour-wheel were not as optimal as those adjacent or opposite? How do you envisage the findings to be used at scale?

Experimental design

- Manuscript seems to be in scope with PeerJ, ethical considerations not applicable.

- Overall M&M also need to be reviewed (minor questions). For instance Figure 2 is mentioned before figure 1; 12h:12h and PFD units are mentioned twice; would be usual to state n number and start size range (presume blotted wet weight) for collected seaweeds - particularly as they are collected from the wild as juveniles with unknown provenance and may show a wide variance; presume also that there was no light pollution between treatments, and the fluorescent light producer and brand needs to be stated (presume wide spectrum type - others exist, e.g. narrower office lighting which can be harsh blue or domestic warmer light).

Main design or concept questions are as follows, bearing in mind that the main purpose of the study is to understand whether additional lighting could work at scale in the sea, to improve seaweed growth.

1) The trial is with juveniles over a relatively short 18 day period at lab scale. Do you think it is possible to reliably inform a future large scale effort?
2) If so, would a better control (or reference) involve normal sunlight, or turbid water? Or perhaps an additional/future field study?
3) Juvenile seaweed is described twice as more sensitive to light (line 240) and culture conditions (line 105) - presumably compared to older specimens. Is the model sufficiently robust to be carried through over 3 different light stages (or indeed seasons mentioned in Figure 6)?
4) The M&M is a little thin regarding how other literature has been used to inform and project the model (equation) over greater time periods and culture conditions. Granted, supplementary material is provided, but it is qualitative and discusses mainly other genera and species. Is it possible to show how additional data was used?

Validity of the findings

See above.

Different LED colours and varying impact on seaweed production is overviewed, sufficient numerical data is provided.

The manuscript would benefit from review, aiming at the research and commercial seeding/nursery sector, as it appears too early a stage or scale to suggest this could be worthwhile to attempt at sea (in terms of results, never mind physically how it could be achieved and whether it is economically viable). An interesting and useful addition could be to investigate the energy and cost of the LED range vs fluorescence, with regard to the results of algal biomass with relation to quantity and quality (composition).

Best of luck.

·

Basic reporting

Manuscript was good throughout. Some minor grammatically errors are address in the general comments section.

Experimental design

The research revolves around the effects of different wavelengths of light on growth of two seaweed cultivars. The implementation of the lighting treatments was innovating and was designed to answer the questions at had.

Validity of the findings

The paper was short and to the point. Overall, the authors concluded that the use of LEDs increased the growth rate compared to fluorescent lighting. Furthermore, the increase in growth was fairly substantial and should be of great interest to the community involved in seaweed production.

Additional comments

Overall, the manuscript was well written. The authors were quick and to the point which is refreshing to see. They generally give enough information without over stepping the importance of their research.

Please find specific requests below:
Line 31: Maintained should be maintaining.
Line 46: ‘i.e. pollutants and nutrients flow into coastal waters’ should be re-worded to make the sentence easier to read.
Line 75: Change photosynthesis with photosynthetic.
Introduction: The use of different lighting qualities (i.e., wavelengths) is interesting. I suggest adding a few lines about how different wavelengths of light are able to penetrate water. Some wavelength can penetrate deeper then others and this may be valuable background information.
Lines 85 and 86: It is more traditional to say 12h of light followed by 12h of darkness.
Line 93: I believe there is supposed to be a reference to figure 1 at this point.
Figure 2: The Y-axis should not be labelled absorbance. Absorbance indicates that you measured how well the seaweed absorbed each wavelength of the lighting treatment. Instead, I suggest the Y-axis be labelled fraction of peak wavelength or something as such.
Line 106: Replace settled with placed.
Line 122: Italicize the a after chlorophyll throughout the manuscript. Change photosynthesis with photosynthetic.
Line 125: Change all absorbances with Chlorophyll a and carotenoid content. Although you are measuring the absorbance with the spectrophotometer, it is more correct to say you’re measuring the content itself.
Line 148-149: Change white-LED light is stimulating seaweed growth with white-LED light stimulated seaweed growth. Also remove ‘as’ before compared.
Line 176: Replace as compared with compared to.
Discussion section Light driven shifts in seaweed growth: Again, I believe there should be some discussion about the light penetration within water between different wavelengths. This may provide further insight into why some treatments worked better than others, specifically when compare both white light treatments (LED vs. fluorescent).
Line 208: Change difference with different.
Line 233: Change interfere the in-door cultivation temperature with interfere with indoor cultivation temperatures.

Reviewer 3 ·

Basic reporting

A clear, concise and well-structured manuscript with few typographical errors.There is sufficient background provided to support the aims of the work.

Literature references are mostly sufficient. However see L222: The authors comment on the similarity of findings of the effect of blue light on Dunaliella salina but should also consider more recent data on the effect of red light - see
Y Xu, PJ Harvey (2019) Red Light Control of β-Carotene Isomerisation to 9-cis β-Carotene and Carotenoid Accumulation in Dunaliella salina Antioxidants 8 (5), 148
Y Xu, PJ Harvey (2019) Carotenoid production by Dunaliella salina under red light Antioxidants 8 (5), 123


Correct Fig 3: gluorescent.

Experimental design

The original primary research is within scope of PeerJ. The research fills an identified knowledge gap with respect to effects of LED light on growth and productivity of two species of seaweed

Validity of the findings

All underlying data have been provided and are robust, statistically sound and controlled.The conclusions are well-stated and limited to supporting results

---

## Round 0.2 · Minor Revisions

I apologize for the delay in the decision. I have looked over the past comments, and your responses, and find the current version to be scientifically acceptable.

However, the English, particularly in the newer and edited parts of your manuscript, still needs a round of thorough revision as PeerJ does not extensively edit at the proof stage. Please ensure that you find a colleague or service to very thoroughly go over your work once more before any resubmission.

·

Basic reporting

Overall the re-submission shows clear evidence of sufficient alterations based on the three reviews and editorial comment.

The manuscript is much clearer and easier to read with additional references. I think the final proof would benefit from a final minor editorial from a native English speaker, really a final polish to remove redundant "the" and "and" throughout.

Line 58, it may add further importance if you state that large scale seaweed culture / IMTA may actually help reduce N pollution.

The graphs and diagrams are relevant and well realised, and the discussion relates to the results.

Thanks for including some reference to the reduced intensity of light, corresponding to depth and varying with different wavelengths. Only minor point is the "Intensity Reservation Rate" (S2) which I've not heard of before. Describing this as a "rate" would imply a change in x/y, i.e. perhaps % intensity reduction/cm in this instance. Perhaps the units need to be altered, or the measurement described differently.

Experimental design

Improved in line with above statements, No comment.

Validity of the findings

Significantly different and non-significant data clearly shown and discussed. Conclusion brief and to the point.

Additional comments

No comments

·

Basic reporting

no comment.

Experimental design

Thank you for emitting the fluorescent light if there was not proper replication. However, this does take away a little bit from the paper. I suggest adding some in text references which can allow for some comparison. Such as preliminary results with fluorescent indicate that LEDs in general allow for __% increase in growth. You don't have to put the data sets back in, but some in text references would acceptable when used in limitation.

Validity of the findings

no comment.

Additional comments

I think the authors have addressed most of my concerns. I did notice a section was deleted which I like. It was related to the difference between brown, green, and red alga species. I thought that brought in a nice comparison. I also think the authors have results which they could strengthen this argument. In figure 3. from about day 8 onwards, S. fusiforme has a higher growth rate than U. pertusa under white, blue, and green light. Why is this? you touch a little bit about species specific effects but I think you could expand that section slightly. In terrestiral plants, it is well known that plants with different canopy architectures utilize different wavelengths of light better or worst depending on plant compactness. Is this the same in seaweed. Admittedly, I'm not a seaweed physiologist, but potentially the morphology of S. fusiforme allows for greater growth from those specific wavelengths?

In figure 2, the authors still have the fluorescent light spectrum which can now be taken out.

---

## Round 0.3 · Minor Revisions

Thank you for your resubmission. Unfortunately, the English still is in need of editing; there are too many small errors and edits needed to accept this work and send it to the production office. Please find a reputable service or experienced colleague for this, and provide the name of the company or person upon resubmission. I do not wish to be overly strict about this, but the English of any scientific paper is the responsibility of the authors.

---

## Round 0.4 · accepted · Accept

Thank you very much for your continuing efforts to improve your work. I am happy to move this into production and look forward to seeing the published version.